# Hyperlipidemia and Cardiovascular Risk in Children and Adolescents

**DOI:** 10.3390/biomedicines11030809

**Published:** 2023-03-07

**Authors:** Francesca Mainieri, Saverio La Bella, Francesco Chiarelli

**Affiliations:** Department of Pediatrics, University of Chieti, 66100 Chieti, Italy

**Keywords:** atherosclerotic cardiovascular disease, cardiovascular risk, hyperlipidemia, dyslipidemia, atherosclerosis, lipid metabolism, children, adolescents

## Abstract

Atherosclerotic cardiovascular disease (ASCVD) represents the major cause of morbidity and mortality worldwide. The onset of the atherosclerosis process occurs during childhood and adolescence, subsequently leading to the onset of cardiovascular disease as young adults. Several cardiovascular risk factors can be identified in children and adolescents; however, hyperlipidemia, in conjunction with the global obesity epidemic, has emerged as the most prevalent, playing a key role in the development of ASCVD. Therefore, screening for hyperlipidemia is strongly recommended to detect high-risk children presenting with these disorders, as these patients deserve more intensive investigation and intervention. Treatment should be initiated as early as possible in order to reduce the risk of future ASCVD. In this review, we will discuss lipid metabolism and hyperlipidemia, focusing on correlations with cardiovascular risk and screening and therapeutic management to reduce or almost completely avoid the development of ASCVD.

## 1. Introduction

Atherosclerotic cardiovascular disease (ASCVD) and its long-term consequences stand out among the main causes of death worldwide [1]. Atherosclerosis is a pathologic process that starts in adolescence, gradually raising the risk of cardiac events, such as heart disease, myocardial infarction (MI), and stroke in later life. Lifestyle variables, medical disorders, including elevated body mass index (BMI), diabetes mellitus (DM), hypertension, the potential tobacco use during late adolescence, as well as genetic and environmental conditions determining an increase in lipid levels, are all risk factors for ASCVD that can be found in children. Specifically, during the last decades, overweight and obesity in children and adolescents have become epidemic, representing one of the major health burdens all over the world. Consequently, an increasing medical focus on the issue of hyperlipidemia has been developed. It has been shown that the increased prevalence of overweight and obesity is associated with elevated serum lipid levels, which is considered one of the main cardiovascular risk factors in childhood [2]. Currently, the prevalence of hyperlipidemia is steadily increasing worldwide; according to data from the National Health and Nutrition Examination Survey (NHANES), 20% of patients aged 12–19 years old show lipid disorders. As shown in several studies, the prevalence varies significantly in relation to the specific country, type of dyslipidemia and children’s body weight. Particularly, the category of children with obesity usually presents an even higher prevalence of hyperlipidemia, up to 42% [3,4,5,6]. Thus, the prevalence of dyslipidemia increases with increasing BMI, but also with increasing age, with 15% of children aged 6–11 years and 25% of adolescents aged 12–19 years presenting at least one adverse lipid level, while no significant differences were found in the frequency of dyslipidemia in boys and girls [7,8]. Children from economically disadvantaged backgrounds may have particular vulnerability [9]. Definitely, the greater number of cardiovascular risk factors, the higher the cardiovascular risk [10,11]. This condition leads to a concomitant increase in the prevalence of cardiovascular disease (CVD) in childhood. Although these cardiovascular risk factors can appear at a young age, they tend to track into adulthood and determine a subsequent high risk for cardiovascular events in adult patients, as recently demonstrated by Jacobs et al. [12]. In this study, a metabolic assessment consisting of BMI, serum lipid levels and systolic blood pressure evaluation was performed on children with an average age of 11.8 years. The results were then reevaluated 35 years later, with the evidence of 3.8% of patients experiencing an ASCVD incident by the age of 46, and 0.8% of these events were fatal. This emphasizes that the length of time that cholesterol is elevated predicts ASCVD occurrences [13]. PubMed and Google Scholar were searched to identify the most recent and/or significant studies regarding lipid metabolism, the pathophysiology of hyperlipidemia and associated cardiovascular risk in children and adolescents, and the crucial role of early screening and targeted therapy, both pharmacological and nonpharmacological in the pediatric population. In fact, the aim of this review is to underline the role of hyperlipidemia that, being associated with a strong cardiovascular risk, may most likely result in the development of ASCVD. Therefore, identifying at-risk pediatric patients is crucial to improve therapeutic management and lifestyle adjustments as early as possible to avoid or at least decrease the occurrence of ASCVD.

## 2. Lipid Classes

Fully understanding how lipids are metabolized in the body is necessary to better comprehend lipid disorders in children. Precise regulation of the equilibrium between energy intake, storage and consumption is necessary for maintaining metabolic homeostasis in humans [14]. Phospholipids, cholesterol, cholesterol esters and triglycerides (TG) represent the four main classes of lipids in the plasma that need to be packaged and transferred [15].

### 2.1. Phospholipids

Phospholipids are composed of a hydrophilic phosphate nonpolar “head,” and a “tail” made up of a hydrophobic fatty acid and are a fundamental part of cell membranes. Phosphatidylcholine and sphingomyelin are classified as polar phospholipids. The enzyme lipoprotein-associated phospholipase A2 (Lp-FLA2), secreted by white blood cells, hydrolyzes the phospholipids of the oxidized low-density lipoprotein (LDL) surface monolayer, resulting in the generation of endogenous inflammatory mediators-lysophosphatidylcholine and oxidized nonesterified fatty acids, which play a significant role in atherogenesis [16]. Thanks to their position on the surface of lipoproteins, phospholipids can affect several cellular processes, including the activation of the innate immune defense [17].

### 2.2. Cholesterol

Cholesterol plays a key role in the synthesis of several molecules, such as steroid hormone, vitamin D, bile acids and cell-membrane integrity, all products with indispensable biological functions for our body. Despite the positive effects of cholesterol, its high levels are toxic, and they undergo firm regulation by several pathways. They are primarily represented by de novo synthesis in the smooth endoplasmic reticulum via 3-hydroxy-3-methylglutaryl-coenzyme-A (HMG-CoA) reductase, LDL receptor-mediated endocytosis, intracellular esterification via the enzyme acyl-coenzyme-A cholesterol acyltransferase (ACAT), and cholesterol efflux from plasma membranes to cholesterol acceptor particles in the high-density class of lipoproteins [18]. All these pathways show a different regulation, by inherited and environmental modulators influencing gene transcription, as for the first two and the last pathway, and by substrate interactions sensitive to membrane cholesterol content for the ACAT pathway. About 30% of the daily cholesterol demand derives from food incoming from the body, while 70% is synthesized in hepatocytes from acetyl-CoA. Mevalonate, which is the result of HMG-CoA reductase influence, bypassing intermediate metabolites, is then converted to cholesterol. Cholesterol is absorbed in the small intestine and contributes to chylomicron formation. Neutral sterols and bile represent the products of cholesterol catabolism [19].

### 2.3. Cholesterol Esters

Cholesterol esters are formed by the esterification of cholesterol with long-chain fatty acids and are synthesized in the liver and intestine by ACAT and in the plasma by lecithin-cholesterol acyltransferase. Cholesterol esters represent the means by which cholesterol is transported through the blood by lipoproteins, along with TG (or triacylglycerol), but also the way cholesterol itself can be stored in the cells. Thus, these molecules have a crucial part in metabolic pathways at the basis of cholesterol trafficking and homeostasis [20].

### 2.4. Triglycerides (TG)

TG are esters derived from glycerol and three fatty acids, synthesized in many organs, such as the liver, intestine, and adipose cells. They are present in the blood to enable the bidirectional transference of adipose fat and blood glucose from the liver. Diet TG transport is realized by chylomicrons. The products of TG cleavage enter the epithelial cell membranes in the small intestine villi during absorption and subsequently are transferred to the smooth endoplasmic reticulum, which is the site of TG resynthesis. TG is usually stored in both visceral and subcutaneous adipose cells; nevertheless, in stressful situations, TG lipolysis occurs, with the following formation of free fatty acids, taken by muscle cells as a substrate for mitochondrial oxidation and ATP synthesis [21]. TG is very useful for cellular signaling but also to produce metabolic energy by providing a highly concentrated source of nonesterified fatty acids (NEFA) upon hydrolysis. However, lipotoxicity may be the result of excessively high NEFA concentrations [22].

## 3. Lipoproteins

Lipoproteins, which are classified according to their relative content of four major lipids and differ significantly in their functions, act as the body’s primary storage reservoir and are essential to this dynamic system [23]. The macromolecular structure of all lipoproteins consists of a hydrophobic cholesterol ester and triglyceride core surrounded by a hydrophilic surface of phospholipid and free cholesterol, with a variable apoprotein content that is essential for receptor binding and function. Human beings have developed a system that allows the solubilization and transportation of hydrophobic energy-rich lipid molecules within lipoproteins from sites of intestinal absorption and hepatic synthesis to and from sites of cellular utilization through the circulatory space. Chylomicrons, very-low-density lipoproteins (VLDL), intermediate-density lipoproteins (IDL), LDL and high-density lipoproteins (HDL) in two subfractions (HDL2 and HDL3), and lipoprotein(a) (Lp(a)) are the main plasma lipoproteins in the sequence of hydrated density increase and size reduction. 

### 3.1. Chylomicrons

ApoB48, apoE and apoCII are the main chylomicron apoproteins. If the first two act by binding the chylomicron remnants to hepatocyte scavenger receptors, apoCII is a cofactor activating a plasma enzyme called lipoprotein lipase (LPL), which realizes the hydrolyzation of lipoproteins with a high TG content. Their metabolism is better explained later in the text (see paragraph 4).

### 3.2. LDL-Cholesterol and VLDL-Cholesterol

The most atherogenic lipoprotein classes are LDL cholesterol (LDL-C) and VLDL cholesterol (VLDL-C), complex protein-lipid supramolecular complexes that transport cholesterol and polyunsaturated fatty acids to cells, respectively. In fact, these lipoprotein particles, as the primary initiators of atherogenesis, alter endothelial characteristics, enhance blood cell adhesion, and activate monocyte/macrophage chemotaxis, resulting in smooth muscle cell proliferation. As shown by Brown and Goldstein, cholesterol transportation into the cell as a part of LDL-C is realized by receptor-mediated endocytosis [24]. Thus, normolipidemia can be maintained through the receptor-mediated capture of LDL-C, which ensures blood cholesterol levels remain in the normal range, avoiding atherosclerosis development. An even higher role in atherogenesis belongs to peroxide-modified LDL-C, which is rapidly recognized and captured by macrophages’ scavenger receptors. As a result, cholesterol esters are accumulated, macrophages are converted into foamy cells, determining degradation of fatty acids with subsequent release of cytotoxicity, monocyte chemotaxis is encouraged, and leukotrienes formation is stimulated [25,26]. Especially in case of dysfunction, the endothelium can be crossed by all apoB-containing lipoproteins, causing the initial deposition of lipids and the development of atherosclerotic plaques [27]. 

### 3.3. HDL-Cholesterol

Generated in the liver and gut, nascent HDL particles are principally made up of phospholipids and apolipoproteins A-I and A-II, and they are able to acquire cholesterol through the action of the ATP binding cassette transporter A1 (ABCA1). HDL particles exert their main function by realizing the reverse cholesterol transportation from peripheral tissues, the vascular wall, to liver cells, gaining the name of anti-atherogenic. Lecithin cholesterol acyltransferase (LCAT) is the enzyme that realizes HDL esterification. In addition, HDL-cholesterol (HDL-C) has a protective role on LDL-C, avoiding their modification towards the atherogenic orientation and stimulating the utilization of TG-rich lipoproteins [28,29]. Only a modest portion of circulating cholesterol is free, as esterification of VLDL, IDL and LDL carried out by cholesterol ester transfer protein (CETP) affects only two-thirds of plasma cholesterol. Mature HDL particles can also derive from the acquisition of several lipoproteins. A great difference between LDL and HDL consists of the induction of atherosclerosis by depositing cholesterol in foam cells realized by LDL, and on the opposite, elimination of cholesterol from these cells from HDL [30]. As demonstrated, low plasma HDL-C concentration represents an independent risk factor for ASCVD; nevertheless, there is no evidence up to now that elevated plasma HDL-C levels are linked to ASCVD risk reduction [31]. However, female gender, estrogens, increased physical activity and weight loss are factors linked to elevated levels of HDL-C. On the opposite, male gender, progestogens, overweight/obesity, hypertriglyceridemia, type 2 diabetes and bad habits, such as alcohol, tobacco use and high carbohydrate consumption, more often determine low HDL-C plasma concentrations [29].

### 3.4. Lipoprotein a 

Lipoprotein a (Lp(a)) is an LDL-C conjugate with a particular apoprotein (a) that shows a similar structure to plasminogen and is able to block the fibrinolysis process. Furthermore, because of the transfer of oxidized phospholipids, Lp(a) exhibits pro-inflammatory characteristics. The higher the plasma levels of Lp(a), the greater the risk of ASCVD; however, in some patients, the magnitude of this risk may be lower than that related to LDL-C. Although, it has been recently found that people presenting Lp(a) levels higher than 180 mg/dL (>430 mmol/L) are at increased ASCVD risk as if they had familial heterozygous hypercholesterolemia [32,33]. As Lp(a) amount is genetically transmitted in almost 90% of cases, it is more likely that an abnormally high level of Lp(a) determines a novel and prevalent hereditary dyslipidemia associated with an increased ASCVD risk throughout life [34].

## 4. Exogenous and Endogenous Metabolism of Lipoproteins

Lipid particles in the plasma are constantly changing due to both enzymatic activity and the migration of lipids and proteins between the particles. Lipid metabolism can be divided into two main pathways: the exogenous lipid metabolism that results from lipoproteins deriving from ingested fat and the endogenous one from lipoproteins synthesized in the liver. As shown in Figure 1, TG deriving from ingestion, immediately after being absorbed by the intestinal mucosa, forms chylomicrons which are released first into the lymphatic system and subsequently into the bloodstream. The union between triglyceride and, to a lesser extent, cholesteryl ester, with apoB48, a truncated version of apoB100, create chylomicrons in the mucosal cells. This specific reaction is mediated by microsomal triglyceride transfer protein (MTTP). Once in the circulation via the thoracic duct, chylomicrons acquire apoCII and apoE, and then the LPL, placed on the surface of endothelium and activated by apoCII, hydrolyses the TG in chylomicrons. As a result, fatty acids and apolipoproteins are removed from the particles, leading to the production of chylomicron remnants, which can be absorbed by the liver. 

The formation of VLDL particles in hepatic cells from triglyceride and MTTP is part of the endogenous pathway, illustrated in Figure 2. This process is similar to that for creating chylomicrons, but in this case, the apolipoprotein is apoB100, and also, apolipoprotein E has a part in forming novel VLDL. Instead, LDL remnants, which can be taken up by the liver, are formed following the loss of fatty acids due to LPL action on VLDL particles. The result of further decomposition is the production of IDL; however, LDL represents the final step of lipase activity on VLDL and IDL. LDL is rich in cholesteryl ester and apo B-100, which can be used as a ligand and allows LDL internalization through specific LDL receptors present on cells [24]. Through the breakdown of LDL, cholesterol is released inside the cells, but high cellular cholesterol levels determine a negative effect on LDL receptor synthesis, a condition in which statins’ mechanism of action can intervene.

## 5. Hyperlipidemia Classification

Hyperlipidemia is the result of increased blood levels of lipids due to disorders in the body’s lipid transport system caused by both genetic and acquired conditions. D.S. Fredrickson et al., in 1967, created a hyperlipidemia classification based on the association of lipid and lipoprotein metabolism abnormalities with ASCVD, approved by WHO experts and consequently used all over the world [35,36]. This classification is based on which lipoprotein class is increased, as visualized on paper electrophoresis and supported by chemical quantitation of plasma TG and cholesterol. The five types are shown in Table 1.

### 5.1. Type I—Hyperchylomicronemia 

Hyperchylomicronemia is characterized by the presence of severe hypertriglyceridemia and fasting excess chylomicrons refractory to conventional treatment due to a hereditary defective removal of these particles related to reduced or absent LPL activity or apoCII deficiency. Type 1 dyslipidemia is considered conditionally atherogenic. The atherosclerotic plaque may be the result of chylomicron residual accumulation in the endothelium because of their prolonged circulation in the blood. Familial chylomicronemia syndrome (FCS) is a very rare genetic hyperlipidemia—1–2 cases/million people, more common in children younger than 10 years, determined by the presence of biallelic pathogenic mutations in the five genes encoding the enzyme and proteins directly implicated in the triglyceride lipolytic cascade. Given the genetic inheritance, FCS usually manifests during infancy; nevertheless, many individuals are diagnosed as adults, and the onset of symptoms as a child is evaluated retrospectively [37]. LPL gene is the most common; however, APOC2, APOA5, GPIHBP1, and LMF1 are also seen to contain causing FCS variants. LPL activity is the shared result of all these genetic alterations that determine a consistent hyperchylomicronemia, known in the literature as type I hyperlipidemia [38]. The typical symptom is abdominal pain, often radiating to the back, the manifestation of acute pancreatitis, potentially fatal [39]. This pain results from lipid micro-occlusions of pancreatic vessels and the massive release of free fatty acids that exert toxic-necrotizing action [40]. Eruptive xanthomas, lipemia retinalis, and/or hepatosplenomegaly have also been described. Sometimes other chylomicronemia syndromes may show similar symptoms as FCS, and this is the case of multifactorial chylomicronemia, lipodystrophy, glycogen storage disease and the presence of autoantibodies against LPL or GPIHBP1 [41,42,43].

### 5.2. Type IIa—Familial Hypercholesterolemia

Familial Hypercholesterolemia (FH) is characterized by elevated cholesterol blood concentrations, specifically markedly high LDL-C levels, that are registered early in life. It represents the dyslipidemia that most causes atherogenic conditions, showing an increased risk of early and progressive ASCVD, which can sometimes be detected even in children and adolescents. Indeed, the clinical manifestations are characterized by cutaneous and tendon xanthomas, which are inevitably associated with accelerated atherosclerosis [44]. FH is an inherited genetic disorder with an autosomal pattern, and according to the different mutations involved, it can be identified as a homozygous (HoFH) or heterozygous (HeFH) entity. All the mutations causing FH are related to the metabolism of LDL; in fact, 85–90% of patients show abnormalities in the LDL receptor (LDLR). Specifically, this leads to a condition of absent or decreased activity of the accumulation of LDL particles in the plasma [45]. Other genes, even less rare, involved in the development of FH encode for ApoB, protein convertase subtilisin/kexin 9 (PCSK9) and LDLRAP1 [46]. HoFH is very rare, approximately occurring one per million individuals worldwide, and is associated with more severe alterations, while HeFH is more frequent, with a prevalence of one affected individual every 311–313 people, and represents the less severe part of this disorder [47,48]. Thus, in HoFH patients, elevated LDL-C levels may be found at birth or even earlier due to detection in utero, and premature clinical manifestations, such as angina pectoris, could occur as early as adolescence, along with first cardiovascular events [49]. Indeed, patients who do not receive treatments meet death earlier than the second decade of life.

### 5.3. Type IIb—Combined Hyperlipidemia

This dyslipidemia is defined by elevated LDL-C associated with elevated VLDL and hypertriglyceridemia. It is an autosomal dominant disorder, and the causative defect is the overproduction of apoB100, associated with the normal catabolism of LDL [2]. Among primary dyslipidemia, familial combined hyperlipidemia often shows this type of dyslipidemia [50]. More commonly, combined hyperlipidemia is secondary to several conditions, and this is linked to type 2 DM, metabolic syndrome and chronic kidney disease [51]. Children may present first symptoms by 5 years of age, and they are represented by xanthomas, cardiopathies and vascular anomalies. It is assumed that these patients have a high risk for CVD; thus, management of this disease is crucial for the prevention of ASCVD.

### 5.4. Type III—Familial Dysbetalipoproteinemia

Patients with type III hyperlipoproteinemia, or Familial Dysbetalipoproteinemia (FD), show elevated levels of TC and TG, usually in the range of 300–500 mg/dL (3.39–5.65 mmol/L), derived from increased plasma levels of chylomicron and VLDL remnants enriched in cholesterol esters and apoE. This chylomicron and VLDL remnants accumulation is caused by E2/E2 homozygosity for the apoE gene. ApoE is a significant component of chylomicron and VLDL remnants and acts as a ligand for receptor-mediated particle absorption by the liver [52,53]. Indeed, mutations related to apoE determine an impaired clearance of remnant lipoproteins by hepatic lipoprotein receptors. ApoE2, apoE3 and apoE4 are the three most common genetic variants. In the case of homozygosity for the E2 form, clearance of chylomicron remnants and VLDL is greatly slowed [54]. As shown by Fredrickson and his colleagues, type III hyperlipoproteinemia has been defined as a highly proatherogenic disorder [35]. Diagnosis emerged from the presence of a wide band on paper electrophoresis and the ultracentrifugation of excessively cholesterol-enriched chylomicron and VLDL remnants particles. Frequently, this disorder manifests after 20 years of age; however, a second “hit” may occur, which may precipitate the hypertriglyceridemia earlier [55].

### 5.5. Type IV and V—Familial Hypertriglyceridemia

Type IV is characterized by an elevated amount of VLDL-C, subsequent to either elevated production or reduced catabolism of VLDL, hypertriglyceridemia and accounts for 45% of the general population. This hyperlipidemia can be a manifestation of hereditary hypertriglyceridemia due to a defect in the genes encoding LPL or its cofactor-apoCII. TG levels for type IV range from 2.3–5.6 mmol/L, and HDL-C tends to be decreased. Typical features, which often appear in adolescence, are represented by hepatomegaly of a solid elastic consistency with a blunt edge that stands for the presence of fat hepatosis, DM, xanthomas, and even rare lipaemia retinalis [56].

Type V is the result of a hereditary defect of the apoCII gene that is expressed by increased TG that exceeds 1000 mg/dL (11.30 mmol/L), VLDL-C and chylomicron levels. Acute pancreatitis can occur as a consequence of TG values of 20 mmol/L. It is rarer, showing a prevalence of 5% of all dyslipidemia cases.

Familial hypertriglyceridemia, characterized by excessive TG synthesis, may manifest as type IV or type V hyperlipoproteinemia. These diseases are inherited in an autosomal dominant pattern and are associated with a family history of recurrent pancreatitis and premature CVD, depending on whether these patients show principal increases in chylomicrons and/or VLDL. The manifestation of familial hypertriglyceridemia is generally full in adult patients; however, high levels of TG are present as early as childhood in patients with familial hypertriglyceridemia. This can be explained by the complicated relations between genetic and environmental factors, including overweight/obesity, diabetes and treatments [55].

## 6. Secondary Hyperlipidemia

Sometimes, instead of being the result of a lipid metabolism disorder, hyperlipidemia can derive from a “nonlipid” condition, however, with consequences entirely similar to those found in the primary lipid forms. Thus, secondary hyperlipidemia is commonly caused by uncontrolled diabetes, hypothyroidism, hepatic and renal failure, and obesity. In addition, it can also be determined by specific drugs such as contraceptives, protease inhibitors, retinoids, corticosteroids and androgenic steroids and a lifestyle sustained by dietary abuse in both quantitative and qualitative terms, exaggerated alcohol consumption, and a sedentary lifestyle [57]. These and other major causes of secondary hyperlipidemia are shown in Table 2. Before starting specific treatment, the nature of the dyslipidemia must be assessed. 

## 7. Variability of Lipid Spectrum

Changes in lipid levels occur together with normal development and maturation, and children of different ages, ethnicities, and genders have different amounts of “normal” cholesterol and lipoproteins. At birth, lipoprotein levels in cord blood are very low and gradually increase throughout the first two years of life [58,59]. After infancy, lipid and lipoprotein levels remain largely steady until adolescence. During puberty, TC and LDL–C levels decline with age before increasing in late adolescence and the third decade of life, respectively [60]. Even if the percentiles of normal lipid levels differ by ethnicity, the risk of atherosclerosis, evaluated by the CIMT, is similarly associated with lipid values and risk factors. Thus, data and percentiles of lipid values by age and gender have been presented uniformly [60,61]. Most epidemiologic studies concur that there is a strong statistical correlation between childhood and adulthood TC and LDL–C values, and it has been shown that approximately 50% of children with lipid levels above the 75th percentile will also have elevated lipid levels in adulthood [62,63,64,65]. Currently, wide population-based data from the Lipid Research Clinical Prevalence Study, which collected fasting lipoprotein profiles from more than 15,000 children until 1976, and the NHANES, which analyzed lipid levels in 7000 children from 1988, are used to calculate normal lipid values in pediatric age groups [66,67,68,69]. Current guidelines and recommendations (fully available at peds_guidelines_full.pdf (nih.gov) [2] accessed on 1 October 2012) from the American Heart Association/American College of Cardiology (AHA/ACC), the National Heart, Lung, and Blood Institute (NHLBI), and the American Academy of Pediatrics (AAP) use cutoff points defining “acceptable,” “borderline,” and “abnormal” lipid levels, even though these thresholds have not been proven to be reliable predictors of CVD outcomes (Table 3) [2,70,71]. 

## 8. Cardiovascular Risk

ASCVD can be the result of several risk factors, which can be divided into modifiable and non-modifiable ones. Among the latter, there are gender, age and genetic predisposition for CVD. For instance, positive family history of CVD or cases of premature manifestation of CVD in the family are strong risk factors. Dyslipidemia, instead, is included among modifiable risk factors, together with arterial hypertension, DM, factors contributing to thrombosis, elevated values of homocysteine, uric acid, inflammation markers, unhealthy lifestyle habits, such as overweight/obesity, low physical activity, consumption of excessive saturated fats and refined carbohydrates [72]. Other conditions associated with early atherosclerosis development, also defined as nontraditional risk factors, are hypothyroidism, HIV infection, systemic lupus erythematosus, congenital heart defects, cardiomyopathy, radically cured neoplasm, organ transplantation, chronic renal diseases, hypersympathicotonia, hypodynamia, adolescent depressive and bipolar disorders [71]. Unfortunately, specific atherosclerotic markers for children and adolescents have not been identified; however, there are several conditions that may influence the atherosclerotic process, in particular, the level of atherogenic and anti-atherogenic lipid and lipoprotein fractions in blood. In fact, as shown in literature, climatic and geographical circumstances, diet, social environment, and a child’s somatotype have a great impact on atherosclerosis and obesity development [73,74]. In the P-Day Study (Pathobiological Determinants of Atherosclerosis in Youth), post-mortem studies found that cardiovascular risk factors such as hypercholesterolemia and tobacco use were related to the amount of fatty streaks and plaques in childhood [75]. A positive link between fatty streaks and LDL-C and a negative association with HDL-C were found in the Bogalusa Heart Study through the analysis of the coronary arteries and aortas of 35 young autopsied subjects [76]. In addition, this study revealed the presence of fatty streaks and fibrous plaques during childhood, with a prevalence of 50% and 8%, respectively, while the prevalence in young adults was 69% and 85%, respectively. These data confirmed that atherosclerosis originates during childhood, even though children with the ongoing atherosclerotic process are usually asymptomatic [77,78].

## 9. Association of Hyperlipidemia with Thrombosis and Fibrinolysis 

Endothelium has a central role in the future onset of the atherosclerotic process. Its anti-inflammatory, anti-thrombotic and pro-fibrinolytic characteristics can be expressed when the endothelium appears intact and healthy, showing a smooth surface able to ensure the regulation of blood flow. The integrity of the endothelium might be disrupted by the presence of hyperlipidemia, which usually causes wall abnormalities, both functionally and structurally. Vascular tone is mediated by the balance of vasodilator and vasoconstrictor factors and the regulation of functional activity of smooth muscle cells, realized by the endothelium [79,80]. Oxidation-sensitive pathways, impacting nitric oxide bioavailability, mediate many of these actions [81]. Atherosclerosis has been shown to be associated with thrombosis; specifically, the link between these two conditions is represented by atherosclerotic plaque rupture or hemorrhage with luminal thrombosis [82,83]. Although children rarely present advanced atherosclerotic lesions, such as acute coronary syndromes, due to the actions of both platelets and clotting factors, it has been demonstrated that, in the presence of dyslipidemia, they express an impaired fibrinolytic activity. This was revealed by Albisetti et al. thanks to a study on 36 asymptomatic children with dyslipidemia, who, compared with 26 control patients with venous occlusion stress testing, had reduced levels of tissue plasminogen activator at baseline [84]. A great increase in plasminogen, alpha-2-macroglobulin, and fibrinogen levels is thought to be a sign of endothelial dysfunction. In line with what has been explained so far, HMG-CoA reductase inhibitors or statins have been demonstrated to be the correct therapy, able to decrease thrombosis formation and ameliorate the fibrinolytic profile in both adult and pediatric patients [85]. Thus, the evidence so far shows the connection between hyperlipidemia, endothelial dysfunction, and anomalies in thrombosis and fibrinolysis.

## 10. Atherosclerosis Development in Childhood and Adolescence

The atherosclerotic process is often referred to as early onset since it might appear as early as the intrauterine period. In fact, recent data have shown that both fetuses and infants may show a coronary thickening of the intima, with a substantial increase in prevalence during childhood and adolescence [86,87]. Thus, atherosclerosis has been considered a huge pediatric health problem. The thickening of the intima is thought to be the first step of atherogenesis, which subsequently leads to the accumulation of lipids, with the following formation of an atherosclerotic plaque [88]. In fact, carotid intimal-medial thickness (CIMT) is considered a surrogate marker of the atherosclerotic process. As demonstrated in the Muscatine Study, elevated blood cholesterol levels during childhood are linked to increased CIMT in young adults [89]. This suggests how crucial the early initiation of statin therapy in children with FH is to prevent atherosclerosis development, as the severity of this pathological condition is strictly associated with the time of onset and duration of hypercholesterolemia [90]. Together with CIMT, other surrogate markers, which should ideally be non-invasive, to detect atherosclerosis have been developed. These are represented by the analysis of endothelial dysfunction, which measures flow-mediated dilation (FMD), and thickening of the vessel walls, by measuring vessel IMT [91]. Carotid artery IMT predicts MI, cerebrovascular accidents, and aneurysms, conditions that usually develop in adulthood. Furthermore, pulse wave velocity (PWV), which evaluates arterial stiffness, computed tomography (CT), which is able to quantify calcium deposits in the coronary arteries, and magnetic resonance imaging (MRI), which measures atheromatous plaques and is preferred to CT because of the risk of exposure to ionizing radiation, are additional non-invasive methods for estimating atherosclerosis [92]. However, the use of the surrogate atherosclerotic markers listed above is not easily accessible in daily hospital practice because of the need for specialized and expensive equipment and qualified personnel. Peripheral arterial tonometry is one of the tests of more clinical utility that have been identified and is for endothelial dysfunction measure, which is considered a potentially reversible characteristic in the development of atherosclerotic process [93]. Specifically, hypercholesterolemia leads to atherogenesis, causing vascular and endothelial shifts. As a result of endothelial-dependent vascular dilatation, platelet adhesion increases, the generation of a plasminogen activator inhibitor is stimulated, while the plasminogen activator is inhibited, the tissue procoagulation factor is provoked, and heparin-sulfate-proteoglycans no longer functions. All these conditions lead to the disruption of endothelium anticoagulation properties, determining blood clot formation. In addition, pro-inflammatory conditions settle in the endothelium [19]. In patients with FH, atherosclerosis affects mainly the aortic root, the ascending and descending aorta, but also peripheral vessels, including femoral and renal arteries. A huge problem of inflammation and fibrosis has been revealed, as valvular stenosis is the result of calcium and cholesterol deposits on the vessels [94]. Given the evidence that increased cholesterol levels promote atherosclerosis, the maintenance of genetically lower cholesterol values since childhood is linked to a lower risk of coronary heart disease, as demonstrated by randomized Mendelian studies. According to estimates, the chance of developing coronary heart disease is 54.5% (95% CI 48.8%–59–5%) for every 1 mmol/L (38.7 mg/dL) decrease in LDL. This represents a threefold higher reduction than that achieved with the use of statins in older patients [95]. As shown before, type II hyperlipidemia is linked to the development of tiny, dense LDL, low HDL and high TG and LDL-C. Particularly, these characteristics are known as the atherogenic lipid triad, emphasizing their significance in atherogenesis development [96]. Hypertriglyceridemia, as one of the main ASCVD predictors, determines potent vasoconstriction by stimulating the hypersecretion of endothelin 1, which exerts a significantly higher effect than norepinephrine and angiotensin II, and by inhibiting nitric oxide synthesis and L-arginine [97]. Even though atherosclerosis starts in the first decade of life, it undergoes substantial changes between the ages of 5 and 34 years. Specifically, the progression of lesions, which varies in males and females and in different arterial segments, is particularly rapid between 15 and 34 years of age, without severe lesions before the age of 34 years. A rapidly increased prevalence of fatty streaks with age has been shown, which then reaches a plateau, while the number of raised lesions showed exponential growth. By the third decade of life, well-developed atherosclerotic changes in coronary arteries are seen, confirming the high rates of coronary heart disease in middle-aged males. The age-related increase in thickness and cellular density reduction in the intima, registered in men but not in women, have demonstrated sex-related differences in response to growth stimuli. Thus, it seems evident how crucial the lines of intervention for cardiovascular prevention in children and adolescents are to avoid the development of severe ASCVD later in life [98,99].

## 11. Screening for Hyperlipidemia

Despite the rarity of manifest cardiovascular damage in children and adolescents, risk factors and risk behaviors that accelerate the development of atherosclerosis begin in infancy, and there is growing evidence that risk reduction may slow disease progression and reduce cardiovascular complications [2]. However, to properly comprehend the timeliness, relevance, and adequacy of a broad population lipid screening, it is essential to understand both the variances and similarities across genders, ages, ethnicities, and risk groups of children and adolescents. According to data from both the United States and other world countries, approximately 20% of pediatric patients have an increase in 1 or more serum lipid values, rising to 40% in pediatric patients with obesity, defined as BMI >95th percentile [100,101,102]. In childhood, increased levels of TC, HDL–C, LDL–C, HDL–C, and TG predict coronary artery calcium (CAC) and CIMT, which are noted precursors of severe atherosclerosis [2,103,104,105,106,107]. In addition, adolescents with elevated TC values had a fivefold increased risk of suffering from CVD events 40 years later compared to healthy young individuals [2,108]. With these presumptions, it should appear extremely simple to comprehend the key role and leading impact that correct screening may play in preventing CVD and in reducing the incidence of cardiovascular mortality in pediatric patients and in adult life. According to the current NHLBI recommendations, two complementary prevention strategies should be pursued to reduce cardiovascular risk: primordial prevention, to avoid the emergence of risk factors in all youngsters, and primary prevention, requiring knowledge of different risk factors from individual screening [2]. It is known that, along with obesity and insulin resistance, hyperlipidemia in youngsters is rising. Indeed, lipid value screening in selected patients should be a standard part of routine pediatric management, with age-based guidelines meant to detect problems at suitable periods relevant to the disease course. However, a good wide-population screening program should have good reproducibility, accuracy, and acceptability and should impact the disease course and long-term outcome [2]. Specifically, serum lipids and lipoproteins are readily detectable in blood samples, economic expenses are easily accessible, accuracy is well-established, the influence on life’s patients is well-known, and the involvement of suitable candidates plays a very key role in the procedure. Nevertheless, the selection of appropriate children and adolescents for lipid screening is complicated, primarily because most hyperlipidemias are clinically asymptomatic and selective screening alone, such as children with a positive family history alone, fails to identify a significant proportion of children with lipid abnormalities. [2,109,110]. Indeed, the “Coronary Artery Risk Detection in Appalachian Communities” (CARDIAC) Project, conducted in 2010 on more than 20,000 fifth-grade children, showed results indicating that using family history to assess the need for cholesterol screening in children would have both missed many mild hyperlipidemias, and failed to discover a significant proportion with probable hereditary hyperlipidemias requiring pharmacological therapy [111]. The authors concluded that universal cholesterol screening would cover all children with severe hyperlipidemia; however, there are presently no data on the cost-effectiveness of pediatric cholesterol screening approaches. Despite these acquirements, a family history of CVD is acknowledged as a significant risk factor, and 50% of men and 25% of women with FH develop clinical CVD by the age of 50 years due to increased LDL–C levels, according to natural history research [112,113]. Nevertheless, there is no standard approach to evaluating CVD family history, which is frequently erroneous and inadequate. However, about 25–55% of children have family histories of early CVD, supporting individual lipid screening [2,114]. This fact has stunning relevance because in RCTs, including older children and adolescents with FH, therapy with statins significantly reduces LDL–C levels and delays the development of atherosclerosis, and CVD, despite the fact that pediatric medication trials have relatively short follow-up periods [115]. In addition, lipid screening should be extended to the whole family since first-degree relatives of children with increased LDL-C values had higher LDL-C levels and a greater incidence of CV events [116]. However, economic assessments are a major factor in screening decisions, as reported in the same recommendations [2]. The most common dyslipidemic profile in children is a mixed pattern linked with obesity, with moderate to severe TG elevation, normal to mild LDL–C elevation, and lowered HDL-C, with an increased risk for accelerated early atherosclerosis and CVD [2,117,118,119,120,121]. Noteworthily, non-HDL–C levels appear to be sensitive for screening, accurately predicting chronic dyslipidemia, atherosclerosis, and future events better than TC, LDL, or HDL values alone [2,117,122]. Being overweight is an important risk factor for hyperlipidemia, with a clear correlation with increased lipid levels, despite a certain variability [123]. In addition, when secondary hyperlipidemia causes that might exacerbate atherosclerosis are diagnosed, children must be tested for dyslipidemia [2,124]. Different screening methods have been proposed to detect hyperlipidemias in children and adolescents: a universal method, a population screening for a specific age group, a selective method, evaluating a specific high-risk population, a cascade method, screening from an index case to family members, a reverse cascade method, evaluating from pediatric affected patients to other members of the family, and child-parent method, from children screening at a specific age to parents [125]. Assessing the pediatric population is recognized as a universal method of screening [126]. Risk factors for atherosclerosis and early ASCVD in childhood have been listed above [2,71].

According to the literature, if a patient has no risk factors, the approach should depend on age:-Under 9 years: no screening;-9–11 years: all children should be assessed for lipid values one time;-12–16 years: screening is not recommended due to changes in lipid levels during puberty.-17–21 years: one evaluation of lipid values during this age range.

If a child or adolescent has 1 or more risk factors, lipid screening should be undertaken when the risk factor is acquired, which is often after the age of 2 years. However, monitoring should continue until the risk factor is present, with lipid value evaluations every 1–3 years, depending on the patient’s unique risk level [2,71]. Individualization is a key factor for correct lipid screening. Indeed, when a single mild risk factor is present, such as smoking exposure without other risk factors or mild obesity, it is permissible to start screening later and less often, while lipid tests should be performed sooner and more often for high-risk children such as those with Kawasaki illness and coronary artery aneurysm [2,71]. Unfortunately, lipid screening in both children and adolescents has not shown consistent effects so far, probably because clinicians and parents of children with abnormal lipid levels do not adhere to screening and follow-up guidelines, probably due to child or parent reluctance and resistance to dietary and lifestyle changes.

## 12. Treatment of Hyperlipidemia

All children and adolescents, especially those with hyperlipidemia, should practice healthful habits. Several guidelines and suggestions for addressing lipid abnormalities in children have been published, but mostly, neither patients nor doctors are adequately aware of them [127]. Current recommendations promote a healthy lifestyle as the primary therapy for pediatric hyperlipidemia, in which parents plays a crucial role. Indeed, in addition to serving as an example of movement, parents play with their children to facilitate the development of required abilities [128,129]. Specifically, therapeutic lifestyle adjustments should include food alterations, regular physical exercise, weight loss, and discontinuation of cigarette consumption in late adolescence. However, despite the fact that the dietary and lifestyle recommendations assessed in the NHLBI guidelines published in 2011 are generally considered still valid and effective, therapeutic options for children with hyperlipidemia have greatly expanded since then [2,127,130]. 

### 12.1. Daily Physical Activity and Sleep Quality

Recent evidence indicates that sedentary behaviors lead to increased blood lipid levels. Indeed, excessive television viewing is associated with higher TG and lower HDL-C, probably due to a decrease in energy consumption. However, the many recognized advantages of physical exercise include better musculoskeletal, mental, behavioral, and cardiovascular health. Particularly, physical activity improves cardiorespiratory fitness, serum glucose levels and insulin sensitivity, blood pressure and lipid profile [129,131]. Furthermore, physical exercise also enhances bone density and improves balance, protecting children and adults from falls and damage [129,132]. Regular physical activity should be pursued by both healthy people and children with hyperlipidemia or obesity to increase HDL-C, decrease TC, TG, and LDL-C, and, most crucially, reduce body fat and improve BMI [2,127,129,133]. Recently, a dose-response association between an increasing number of minutes of physical activity and improved lipid concentrations (HDL-C and TG values) has been revealed, highlighting once again that at least 60 min of moderate-to-vigorous physical exercise per day, as well as muscle- and bone-strengthening activities at least 3 days per week, should be strongly encouraged in all patients aged 6 through 17 years [2,127,133,134]. Meanwhile, according to the 2018 Physical Activity Guidelines Advisory Committee, children enrolled in preschool between the ages of 3 and 5 years should participate in active play at different intervals throughout the day [134]. However, pediatricians should promote physical literacy and exercise in children and progress toward suggested recommendations by screening and recording gross motor abilities and physical activity during health care visits, discussing the advantages of physical exercise on growth and development, promoting physical exercise for all children via health care, insurance, schools, and community groups [129]. However, up to 80% of adults and adolescents in the United States are considered inadequately active, and much work needs to be done to improve public awareness and the application of guidelines [134]. In addition, parents should encourage their children to get adequate sleep and limit their screen time, such as cell phones and computers, to less than 2 h per day. As revealed by recent meta-analyses, shorter sleep durations are associated with an increased risk of overweight and obesity in children less than 18 years old [135]. According to the current Consensus Statement of the American Academy of Sleep Medicine, infants aged 4–12 months should sleep 12–16 h per 24 h, while toddlers aged 1–2 years should sleep 11–14 h per day to maintain optimal health. Children of 3–5 years of age should regularly sleep 10–13 h per 24 h, and those 6–12 years old should routinely sleep 9–12 h per day. Lastly, teenagers should continuously sleep 8–10 h every day [136]. 

### 12.2. Dietary Modifications

Dietary recommendations play a significant role in reducing cardiovascular risk in both adults and children [127]. Growing evidence shows that a correct dietary approach can significantly improve lipid profile abnormalities. Indeed, the Mediterranean diet may reduce TG, TC, and LDL-C levels in the blood, and improve the CIMT, significantly lowering atherosclerosis progression and cardiovascular risk [137,138,139]. Generally, a healthy diet should reduce the consumption of saturated fat and cholesterol while favoring fiber consumption instead [127,137]. However, there are two different main strategies to achieve effective dietary modifications: an individual approach and a population approach [127,140]. The individual approach consists of a two-step nutritional strategy, depending on the patient. Indeed, the first approach to low cardiovascular risk in children should be the Cardiovascular Health Integrated Lifestyle Diet-1 (CHILD-1), which shares the same recommendations as the population strategy, which consists of generic guidelines for all children with the goal of stopping the development of atherosclerosis and reduce blood lipid levels, as represented in Table 4 [114,140,141]. 

More severe restrictions are required for children with proven dyslipidemia and patients at higher cardiovascular risk with insufficient improvements after three months on the CHILD-1 diet. In fact, according to the Cardiovascular Health Integrated Lifestyle Diet-2 (CHILD-2) recommendations, saturated fat consumption must be limited to 7%, and daily TC intake must be reduced to a maximum of 200 mg per day (Table 4) [114,140,141]. Both CHILD-1 and CHILD-2 are encouraged in children over 2 years of age so as not to restrict calorie intake in infants and young children [127]. However, the diet should always be balanced, allowing children an adequate supply of nutrients and making sure that 50–60% of total daily calories should come from carbohydrates while 10–20% come from proteins [141,142]. Other specific recommendations should be pursued in high cardiovascular-risk children, such as increasing omega-3 fatty acids in the diet of children with hypertriglyceridemia and promoting plant sterol consumption in children with elevated blood LDL-C levels [127]. In addition, non-nutritive sources of calories, such as fast foods, should be limited [141]. However, full evidence-based recommendations for diet and nutrition in children and the estimated calorie needs per day by age, gender, and physical activity level are fully available in the current NHLBI guidelines [2]. All these recommendations strictly require parental collaboration to reduce the risk of premature ASCVD.

### 12.3. Pharmacological Treatment

Adopting healthy behaviors, such as a proper dietary approach, reducing excess weight, daily physical activity, appropriate sleep, and minimizing sedentary habits, often result in favorable outcomes. However, hereditary hyperlipidemias often need stronger dietary restrictions and pharmaceutical treatment due to the ineffectiveness of healthy lifestyles alone [93,127,143]. To date, statins are considered the first therapeutic approach due to their efficacy, safety, and cheapness [127]. These drugs are inhibitors of the enzyme HMG-CoA reductase, the primary player in hepatic cholesterol synthesis. Statins have been proven to reduce several blood lipids, namely by acting on TG, TC, and LDL-C blood levels while also increasing HDL-C [144]. Currently, seven commercially available statins have been approved for children and adolescents. Rosuvastatin is approved for use in children from the age of 7 years, pravastatin and pitavastatin are useful in patients with 8 years or older, while simvastatin, atorvastatin, lovastatin and fluvastatin may be used in patients from the age of 10 years [145]. Notably, pravastatin, simvastatin, and lovastatin are natural compounds, while fluvastatin, rosuvastatin, pitavastatin, and atorvastatin are synthetic products with linked fluorophenyl groups and larger hydrophobic regions [146]. According to the literature, statins may lower LDL-C levels by up to 33% with a dose-dependent mechanism while regulating other lipids less significantly [144]. In addition, their administration decreases the risk of ASCVD in children with both FH and secondary hyperlipidemia, even if lipid levels are not substantially controlled [147]. Given the rarity of side effects, most patients tolerate statins well and have good long-term adherence [127]. Nonetheless, liver transaminase and creatine kinase levels, as well as pubertal development and growth velocity, should be frequently evaluated throughout therapy [127]. Starting dosages of statins should be minimal and progressively raised to the patient’s optimal level, with more attention on children with chronic kidney disease [127]. Lastly, macrolides and antifungal azoles should be avoided in statin-treated patients owing to their interactions with cytochrome P450, as well as during breastfeeding and pregnancy due to their potential teratogenicity [130,148].

Statins are approved by the Food and Drug Administration (FDA) for both heterozygous and homozygous FH [127,145]. In heterozygous forms not well controlled by physical activity and good dietary habits, statins may be useful for maintaining low LDL-C blood levels and reducing the risk of ASCVD. Indeed, LDL-C values in moderate-risk children, mainly represented by heterozygous FH, should be maintained at <130 mg/dL [145,149,150]. In high-risk patients, represented by homozygous FH, LDL-C values should remain < 100 mg/dL or be reduced by 50% from baseline values [145,149,150]. However, up to 80% of homozygous FH children do not achieve LDL-C target values with statins alone [145]. Indeed, to achieve the LDL-C treatment goal, children who maintain high blood lipid levels despite increased statin dose or patients who do not tolerate statins well may require additional cholesterol-lowering drugs. [127,151]. Ezetimibe is the second lipid-lowering drug to be well-tolerated and widely used in children with hyperlipidemia. Its mechanism of action consists in binding the Niemann-Pick C1-Like 1 (NPC1L1) receptor, which mediates cholesterol absorption in gastrointestinal tract epithelial cells. Ezetimibe is frequently used in combination with statins to achieve greater cholesterol control, with LDL-C levels reduced by up to 15% [127,152]. Nonetheless, even when administered alone, ezetimibe consistently reduces LDL-C and TC levels by 27% and 21%, respectively [153]. This drug is typically well tolerated, and transient diarrhea is its most frequently reported adverse effect [152,153]. In children with hyperlipidemia, bile acid sequestrants represent, currently, the last therapy option [127]. Noteworthily, prior to the popularity of statins, bile acid-binding resins such as colesevelam (the only one approved by the FDA in FH children over 10 years) were recognized as crucial therapeutic choices for children [127]. These molecules are not absorbed from the gut, thereby binding bile salts, and lowering cholesterol intake, driving the liver to convert cholesterol into additional bile salt synthesis and subsequently decreasing blood LDL-C levels by as much as 15% [154,155]. Therefore, bile acid sequestrants are associated with significant adverse effects, including nausea, diarrhea, and vomiting, as well as an unpleasant taste, resulting in low treatment adherence in children [156]. Numerous other lipid-lowering medications, including fibrates, vitamin B3, evinacumab, evolocumab, and others, are being evaluated for pediatric use, with encouraging opinions [127]. Specifically, at present, there is no pharmaceutical therapy for familial chylomicronemia syndrome, and primary care choices consist in adopting a very limited, low-fat diet and avoiding the intake of specific drugs and alcohol [157]. However, studies are focusing on two key proteins called apolipoprotein CIII (apo CIII) and angiopoietin-like3 (ANGPTL-3) [158,159,160]. Although ANGPLT-3 inhibitors have not yet been studied enough in affected patients, these drugs have shown proven efficacy in polygenic forms [158,160]. Current drugs targeting apo CIII have been observed to reduce triglyceride levels in familial chylomicronemia syndrome, becoming an available treatment for adult patients. Furthermore, microsomal triglyceride transfer protein (MTTP) inhibitors could represent a good choice in the future [158,159,160]. TG levels in combined hyperlipidemia should be less than 150 mg/dL [161]. The better strategy to achieve this therapeutic goal remains a healthy lifestyle with regular physical activity, especially in patients with obesity [161]. Although statins are currently approved for FH, their efficacy in reducing TG levels in young people is known, decreasing TG levels by 10–15%. Indeed, statins may be beneficial in children with very high TG levels (200–499 mg/dL) and non-HDL-C greater than 145 mg/dL following a 6-month trial of lifestyle improvement and dietary management [161]. In addition, studies are focusing on several novel drugs aiming to reduce TG levels, including microsomal triglyceride transfer protein (MTP) inhibitors, selective peroxisome proliferator-activated receptor (PPAR) modulators, diacylglycerol acyltransferase (DGAT) inhibitors, and ANGPTL inhibitors [161]. 

Nonetheless, healthy lifestyle habits such as a correct dietary approach, excessive weight loss, regular physical activity, and statins are the primary actors in countering hyperlipidemia in children, both for their effective control of lipid levels in most patients and for their positive impact on the natural history of atherosclerosis and, consequently, their capacity to prevent early ASCVD.

## 13. Conclusions

Hyperlipidemia represents one of the major cardiovascular risk factors that affect children and adolescents early in life. Its prevention, screening, and lifelong, personalized management should be started early in childhood in order to achieve the ultimate goal of the best perspectives for children’s future, free from the development of ASCVD. However, there are still potential gaps in care; thus, further studies and specific investigations are mandatory to increase awareness for better and individualized screening and treatment guidelines for pediatric hyperlipidemia.

## Figures and Tables

**Figure 1 biomedicines-11-00809-f001:**
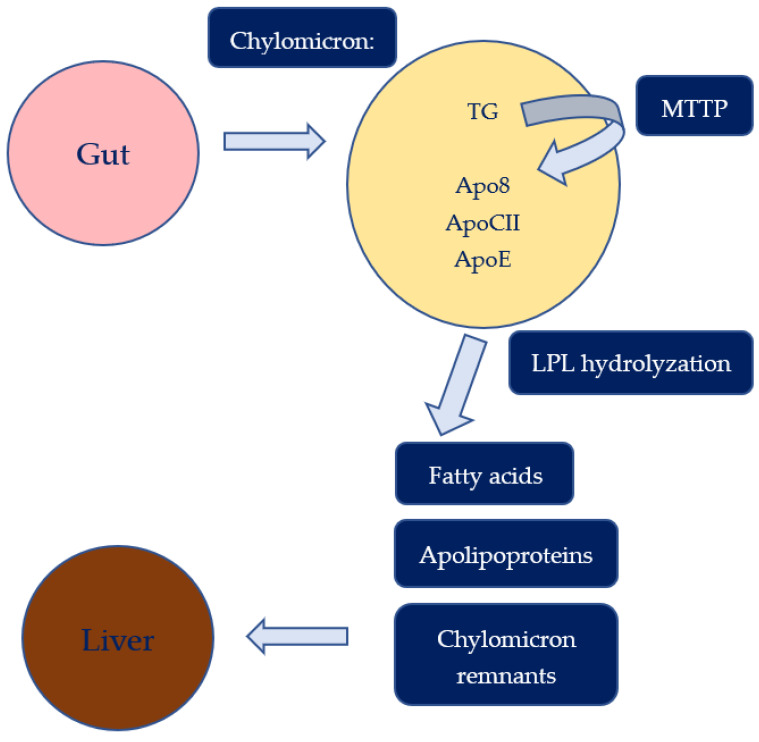
Exogenous pathway of lipid metabolism. Triglycerides (TG), lipoprotein lipase (LPL), microsomal triglyceride transfer protein (MTTP).

**Figure 2 biomedicines-11-00809-f002:**
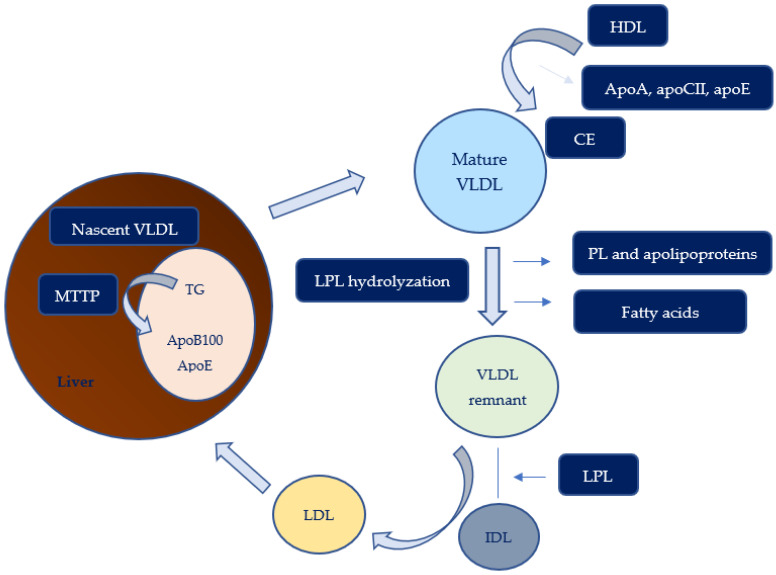
Endogenous pathway of lipid metabolism. Very-low-density lipoprotein (VLDL), low-density lipoprotein (LDL), high-density lipoprotein (HDL), intermediate-density lipoprotein (IDL), lipoprotein lipase (LPL), microsomal triglyceride transfer protein (MTTP), cholesteryl ester (CE).

**Table 1 biomedicines-11-00809-t001:** Fredrickson’s Classification of Hyperlipidemia and association with risk of cardiovascular disease and pancreatitis.

Fenotype	Excess Lipoproteins	Disease	CV Risk	Pancreatitis
I	Chylomicrons	Familial chylomicronemia syndrome	+	++
IIa	LDL	Familial hypercholesterolemia	++++	-
IIb	LDL + VLDL	Combined hyperlipidemia	++	-
III	Beta-VLDL (IDL)	Familial dysbetalipoproteinemia	++	+
IV	VLDL	Familial hypertriglyceridemia	+	++
V	VLDL + chylomicrons	Familial hypertriglyceridemia	+	++

Low-density lipoprotein (LDL), Very-low-density lipoprotein (VLDL), intermediate-density lipoprotein (IDL), cardiovascular (CV) risk. + = present, ++ = high, ++++ = very high, - = absent

**Table 2 biomedicines-11-00809-t002:** Major causes of secondary hyperlipidemia.

Main Causes of Secondary Hyperlipidemia	
Dietetic factors	Diet high in saturated fatty acids and carbohydrates Exaggerated alcohol consumption Sedentary lifestyle
Endocrine-metabolic disease	Type 1 and type 2 DM Obesity Hypothyroidism Hypopituitarism Cushing syndrome Acromegaly Polycystic ovary syndrome Lipodystrophy Pregnancy
Kidney disease	Nephrotic syndrome Glomerulonephritis Chronic kidney disease
Hepatic disease	Alcoholic hepatitis Biliary cirrhosis Cholestasis
Infectious	Acute viral or bacterial infection Human immunodeficiency virus Hepatitis
Inflammatory disease	Systemic lupus erythematosus Juvenile rheumatoid arthritis
Medications	Thiazide diuretics Beta-blockers Immunosuppressants Corticosteroids Oral contraceptives
Other	Kawasaki disease Anorexia nervosa Solid organ transplantation Childhood cancer survivor

**Table 3 biomedicines-11-00809-t003:** Plasma lipid values in pediatric age and related percentiles. Adapted by NLHBI Integrated Guidelines for Cardiovascular Health and Risk Reduction in Children and Adolescents (peds_guidelines_full.pdf (nih.gov)).

Category		Acceptable	Borderline	High
TC		<170	170–199	≥200
Non-HDL-C		<120	120–144	≥145
LDL-C		<110	110–129	≥130
TG	10–19 years	<75	75–99	≥100
	<9 years	<90	90–129	≥130
Category		Acceptable	Borderline	Low
HDL-C		>45	40–45	<40

Values given are in mg/dL. Acceptable <75th percentile, Borderline 75–95th percentile, High >95th percentile. Total cholesterol (TC), high-density lipoprotein cholesterol (HDL-C), low-density lipoprotein cholesterol (LDL-C), and triglycerides (TG).

**Table 4 biomedicines-11-00809-t004:** CHILD-1 and CHILD-2 dietary recommendations by the National Cholesterol Education Program (NCEP) [2,114].

Recommendations	Whole Pediatric Population (CHILD-1)	CHILD-2
Total fat consumption should be limited to:	20–30% of total calorie intake	20–30% of total calorie intake
Saturated fat intake should be less than:	10% of total calories	7% of total calories
The average daily intake of TC should not exceed:	300 mg	200 mg
Daily calories from monounsaturated fatty acids should not exceed:	10–15%	10–15%
Polyunsaturated fatty acids should account for:	up to 10%	up to 10%
It is recommended to take additional dietary fibers (fruits, vegetables, and whole grains such as rolled oats, whole corn, and buckwheat):	5 or more times a day	5 or more times a day
Trans fatty acids should be avoided by children (<1%) in favor of polyunsaturated fatty acids and monounsaturated fatty acids		

## Data Availability

Not applicable.

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
