# Peer review of "Hyperlipidemia and Cardiovascular Risk in Children and Adolescents"

_biomedicines, 2023, doi:10.3390/biomedicines11030809_

Round 1

Reviewer 1 Report

Overall, this is a well written review publication and offers a good overview of the role of hyperlipidemia in ASCVD development. However, before publication some points need to be clarified.

My comments:

Line 46 – Please add short methodology of this review.

Line 97, 102 - From histological point of view there are only four kinds of tissues: epithelial, connective, muscular and nervous. Therefore, such term as “adipose tissue” or “peripheral tissues”, “tissue plasminogen” are not justified.

Line 312, 367 – Diabetes mellitus was already abbreviated to DM.

Line 351 – the term “expression” should be used in relation to genes only.

Line 450 – please use either “(HMG) CoA” (line 450) or “HMG-CoA” (line 45) abbreviation.

Line 728 – there is no such structure as “intestinal tract”.

Line 777 – Abbreviations should be at the beginning. If the authors decide to introduce the whole chapter wit abbreviations than the additional abbreviations in the text are not necessary.

Author Response

Thank you very much for your kind and detailed revision, as it represents a precious opportunity to ameliorate our manuscript with helpful and valuable suggestions.

Response to reviewer 1:

  • A short methodology has been added (lines 82-87).
  • “Adipose tissue” has been changed with “adipose cells” (lines 138, 143).
  • DM abbreviation has been correctly used in lines 361, 390, 416.
  • “Expression” has been changed with “manifestation” in line 400.
  • HMG-CoA is the correct abbreviation present in the text, also in lines 497-498.
  • “Intestinal tract” has been changed with “gut” in line 793.
  • Abbreviations paragraph has been moved at the beginning of the manuscript (lines 22-45) as suggested, thus due to the several, and sometimes difficult, abbreviations, we preferred to leave the first explanation of the abbreviations in the text.

Thank you very much.

Reviewer 2 Report

I have carefully read the narrative review by Mainieri et al., that is interesting, overall well written and update. The only limitation of this review is related to the fact that all the part not related to treatment seems to be only mildly focused on children/adolescents but to lipids/lipoproteins physiology and pathophysiology in humans, independently from the children. I think the text could be more focused, trying to report more data on epidemiology and the age of unveiling, beyond some specific characteristics that dyslipidamia could have in children and adolescents compared to adults, when knwon. Probably a mention to ethnicity and gender differences could be also shortly added.

Author Response

Thank you very much for your kind and detailed revision, as it represents a precious opportunity to ameliorate our manuscript with helpful and valuable suggestions.

Response to reviewer 2:

As kindly suggested, additional data on the pediatric population (epidemiology, pathophysiology, ethnicity, gender and differences with the adult population) have been reported along the manuscript, in the following lines:

  • Lines 61-72
  • Lines 540-546
  • Lines 553-565

Thank you very much.